# Integrating AI and ML in Myelodysplastic Syndrome Diagnosis: State-of-the-Art and Future Prospects

**DOI:** 10.3390/cancers16010065

**Published:** 2023-12-22

**Authors:** Amgad Mohamed Elshoeibi, Ahmed Badr, Basel Elsayed, Omar Metwally, Raghad Elshoeibi, Mohamed Ragab Elhadary, Ahmed Elshoeibi, Mohamed Amro Attya, Fatima Khadadah, Awni Alshurafa, Ahmad Alhuraiji, Mohamed Yassin

**Affiliations:** 1College of Medicine, QU Health, Qatar University, Doha 2713, Qatar; 2College of Medicine, Mansoura University, Mansoura 35516, Egypt; 3School of Medicine, Newgiza University, Giza 12577, Egypt; 4Faculty of Medicine, Alexandria University, Alexandria 21544, Egypt; 5Kuwait Cancer Centre, Sabah Medical Region, Shuwaikh 1031, Kuwait; 6Hematology Section, Medical Oncology, National Center for Cancer Care and Research (NCCCR), Hamad Medical Corporation, Doha 3050, Qatar

**Keywords:** myelodysplastic syndrome diagnosis, artificial intelligence, machine learning, bone marrow smears, peripheral blood smears, flow cytometry

## Abstract

**Simple Summary:**

This paper aims to highlight the latest advancements in the application of artificial intelligence in the diagnosis of myelodysplastic syndrome. This research focuses on a group of blood disorders called Myelodysplastic Syndrome (MDS), which can potentially develop into a more severe condition called Acute Myeloid Leukemia (AML). Detecting MDS early is crucial, but the current methods are time-consuming and labor-intensive. We aim to explore how artificial intelligence (AI) and machine learning (ML) can make the diagnosis of MDS faster and more accurate. AI involves computer programs that can think like humans, and ML is a part of AI that helps computers learn patterns and make predictions. By using these technologies, doctors can improve how they diagnose MDS, leading to better treatment and outcomes for patients.

**Abstract:**

Myelodysplastic syndrome (MDS) is composed of diverse hematological malignancies caused by dysfunctional stem cells, leading to abnormal hematopoiesis and cytopenia. Approximately 30% of MDS cases progress to acute myeloid leukemia (AML), a more aggressive disease. Early detection is crucial to intervene before MDS progresses to AML. The current diagnostic process for MDS involves analyzing peripheral blood smear (PBS), bone marrow sample (BMS), and flow cytometry (FC) data, along with clinical patient information, which is labor-intensive and time-consuming. Recent advancements in machine learning offer an opportunity for faster, automated, and accurate diagnosis of MDS. In this review, we aim to provide an overview of the current applications of AI in the diagnosis of MDS and highlight their advantages, disadvantages, and performance metrics.

## 1. Introduction

Myelodysplastic syndrome (MDS) is a diverse group of hematological malignancies characterized by dysfunctional pluripotent stem cells that fail to undergo proper hematopoiesis and maturation within the bone marrow. Consequently, this leads to an excessive production of immature cells and dysplastic changes in the bone marrow. This disruption in stem cell activity results in a reduction in the formation of healthy blood cells, which manifests as cytopenia in one or more cell types, such as thrombocytopenia, erythrocytopenia, or leukocytopenia [1]. While the majority of adult MDS cases have no known etiology (primary or idiopathic), a small percentage of cases might be linked to an underlying illness (secondary), some of which are linked to autoinflammatory conditions termed VEXA syndrome [2,3]. This illness predominantly affects the elderly and usually has a gradual clinical course [4]. Patients’ presentation typically depends on the manifested cytopenia. They may develop anemia-related symptoms such as fatigue, weakness, and pallor. Recurrent infections and petechial bleeding may also develop as a result of a low number of functional leukocytes and platelets [5,6,7,8]. To establish a diagnosis of MDS, blood tests, a bone marrow biopsy, and genetic analysis are necessary. The diagnosis of MDS requires persistent cytopenia that cannot be explained by any other drug or cause, the presence of < 20% blasts on peripheral blood (PB) or bone marrow biopsy (BM) along with cytogenetic/molecular features (such as mutated *SF3B1*), or the presence of dysplastic morphology greater than 10% in a specific hematopoietic lineage without another explainable cause [9].

It is important to note that approximately 30% of patients with MDS will eventually develop acute myeloid leukemia (AML), which is more aggressive [10]. Hence, early diagnosis and treatment of MDS are crucial to improving patient outcomes [11]. MDS is a complex medical condition that can benefit from advancements in artificial intelligence (AI) and machine learning (ML). AI refers to the development of computer programs that emulate human intelligence. In healthcare, AI has the potential to improve the diagnosis, early detection, prognostication, and monitoring of diseases. Machine learning, a subset of AI, plays a crucial role in harnessing the power of datasets to recognize patterns and generate predictions. What sets ML algorithms apart is their ability to analyze both linear and nonlinear variables simultaneously, enabling them to identify complex patterns and make highly accurate predictions [12,13,14,15]. With the integration of AI and ML, healthcare providers can enhance the accuracy and efficiency of diagnosing MDS. The early diagnosis of MDS can lead to more informed decisions and early treatment plans for patients, leading to improved outcomes and better patient care.

In this review, we aim to summarize the current state of the use of AI in the diagnosis of MDS. We will be discussing the advantages and disadvantages of various ML models and reporting their performance matrices.

## 2. Materials and Methods

To develop our comprehensive search strategy, we employed a combination of medical subject heading (MeSH) terms from PubMed and relevant terms in article titles and abstracts. For our specific disease of interest, MDS, we included terms such as “myelodysplastic syndrome”, “preleukemia”, “MDS”, “myelodysplasia”, and other related terms to ensure inclusiveness. To ensure that our search covered articles discussing the application of AI in MDS, we also incorporated terms related to ML such as “artificial intelligence”, “machine learning”, “AI”, and “deep learning”. This initial search was not limited by language or timeframe. We utilized a polyglot translator to adapt the initial search strategy to Embase, Web of Science, and Scopus [16].

All the studies identified through the search strategy were organized in EndNote, where duplicates were systematically removed. Subsequently, the remaining studies were imported into Rayyan, a screening tool, to eliminate any remaining duplicates and initiate the screening process [17]. It is worth noting that this methodology mirrors the approach we employed in our previous article on thrombocytopenia [18]. By employing this rigorous search methodology, we aimed to ensure a comprehensive and unbiased selection of relevant studies for our review. This review focused on original full-text research articles that specifically explore the application of ML algorithms in the diagnosis of MDS among human subjects. To maintain the study’s relevance and scope, certain studies were excluded based on the following criteria: (1) studies conducted on animals; (2) reviews or non-original articles; (3) conference abstracts; and (4) articles not written in English.

The collected data in this paper include various aspects such as the type of study, publication year, assessed outcomes, methods used to create models, specific ML models employed, evaluation metrics for the models (including sensitivity (SEN), specificity (SPE), accuracy (ACC), and area under the receiver operating curve (AUC)), strengths, and limitations. The AUC values for the models were categorized into different performance levels: unsatisfactory (<0.6), satisfactory (0.6 to <0.7), good (0.7 to <0.8), very good (0.8 to <0.9), and excellent (0.9 to 1.0). In cases where multiple models were utilized within a study, we extracted the metrics for the best-performing model(s). By adhering to these guidelines, we aimed to ensure a thorough analysis of the included studies and provide meaningful insights into the application of ML algorithms in MDS diagnosis.

## 3. Results

The initial search strategy yielded a total of 313 articles from the three databases. These articles were imported into EndNote, where 116 duplicate articles were automatically identified and removed. Subsequently, the remaining articles were transferred to Rayyan, where an additional 19 duplicates were manually identified and excluded. The inclusion–exclusion process was conducted within Rayyan. A total of 178 articles were eligible for screening, of which 29 conference abstracts were excluded, and another 117 articles were excluded due to reporting outcomes irrelevant to our study. Sixteen review articles and 4 non-English articles were excluded. In total, 12 articles met all the inclusion criteria and were included in the final review. A schematic representation of the identification, screening, and inclusion processes is provided in Figure 1, illustrating the flow of articles throughout the review process. Table 1 summarizes the aim of each study and the main advantages and disadvantages of their ML models. Table 2 summarizes the data sources and performance metrics of the best-performing ML models utilized.

### 3.1. Diagnosis of MDS Using BM Samples

BM smears are considered a prerequisite for the diagnosis of MDS. They provide a comprehensive view of cellular composition, morphology, and cytogenetics. The hallmarks of MDS on BM smears include dysplasia and elevated blasts that are <20%. The diagnosis of MDS with dysplasia is only possible when dysplasia reaches 10% in at least one lineage [31]. However, the analysis of BM samples for dysplasia and blasts, along with their quantification, can be difficult and time-consuming for pathologists, which can occasionally lead to the oversight of critical findings. Moreover, the assessment of dysplasia is subjective. Operators have to undergo years of training in order to become competent in the assessment of BM samples, and even then, inter- and intra-variations are present amongst experienced hematologists [32,33,34]. Herein lies the potential for AI to revolutionize MDS diagnosis. By harnessing AI’s capacity for rapid pattern recognition and data analysis, many challenges posed by manual examination of bone marrow samples can be mitigated.

To address the issue of identifying dysplasia, Lee and colleagues presented a convolutional neural network (CNN)-derived ML model that automatically detects dysplasia from images of bone marrow aspirates. The investigators acquired BM aspirates from 34 patients diagnosed with MDS and 24 patients without MDS. They manually captured images within well-spread areas containing nucleated cells to use as examples for the program. In order for the model to function, it had to be able to identify the cells and then classify them. For this, the researchers labeled the boundaries of 946 cells and classified 8065 cells into eight types (normal erythrocytes, normal granulocytes, normal megakaryocytes, dysplastic erythrocytes, dysplastic granulocytes, dysplastic megakaryocytes, blasts, and others). This was used to help train the model to identify and classify these cell types. Eighty percent of the cell images were used for training, 10% were used for testing, and 10% were used for validation. The models created showed excellent AUC for the detection of dysplasia in each cell type, with the AUC ranging from 0.945 to 0.996 [20]. The details for each cell type can be found in Table 2.

The model proposed by Lee and colleagues demonstrated excellent ability in identifying the presence of dysplastic cells in three different lineages, but it is important to note that this model is not able to quantify the percentage of dysplasia. This makes it an excellent auxiliary tool to assist hematologists in recognizing dysplasia when attempting to diagnose MDS. Although the model by Lee was validated by competing with hematologists, it was not externally validated. This form of verification only provides insight into the quality of the model’s prediction and not its generalizability to other samples. Moreover, the model was not trained to distinguish specific changes in different cell types within the BM. Instead, it relied on having an adequate number of normal cells to make accurate predictions about abnormal ones. Furthermore, it is important to note that the study solely evaluated the algorithm’s performance in identifying dysplastic cells without assessing its capability to accurately diagnose MDS [20].

Another model for the detection of dysplasia was proposed by Mori, J. et al. [24] Similar to the one by Lee, the model utilized images of BM smears from MDS and non-MDS patients with labeling performed by morphologists to assist the training of the model. However, Mori’s model utilized decreased granules (DGs) as a marker of dysplasia in granulopoiesis. They classified dysplasia on a 4-point scale, with 0 being normal, 1 intermediate, 2 dysplasia, and 3 severe dysplasia (i.e., severely decreased granules). A total of 1797 labeled images were obtained, with morphologists identifying 134 DGs categorized as DG1 (46), DG2 (77), and DG3 (11). When considering DG1–3 as positive, the classifier demonstrated an AUC of 0.944, ACC of 0.972, SEN of 0.910, and SPE of 0.977. However, since DG1 is vague and ideally the model should be able to identify obvious dysplasia, the researchers excluded the DG1 labels from the analysis and classified the DG1 samples as DG0 or DG2. This yielded an AUC of 0.921, ACC of 0.982, SEN of 0.852, and SPE of 0.989 [21].

The notable distinction of the model presented by Mori, J. et al. is that it relies on cellular features (granules) to detect dysplasia, unlike the model presented by Lee and colleagues. Moreover, the model classifies dysplasia by severity, not just dysplastic vs. non-dysplastic, which can be clinically useful. However, this model was neither externally validated nor challenged by hematologists. Nevertheless, the researchers proposed a “doctor in the loop” strategy (where the expert is supplied with the information acquired) to help limit the number of mistakes made by the model. Another issue was that the number of samples used for the training of the model was small [21].

To address the issue of detecting blasts and quantifying them, Wu, Y. and colleagues presented an AI model that can detect and quantify blasts. BM smears were taken from patients with various hematological conditions. They were divided into a training sample (42), a testing sample (70), and a competition sample (10). Over seventeen thousand images of cells captured by hematologists from the training set were labeled and classified into one of seven cell categories (erythroid, blasts, myeloid, lymphoid, plasma cells, monocyte, and megakaryocyte) by three independent hematologists. If three hematologists could not agree on a cell’s type, they marked it as “unable to identify”. This information was used to train the CNN model to identify and categorize these cell types. To evaluate how well the CNN model performed compared to hematologists, a human-machine competition was conducted involving six visiting staff members. These staff members analyzed the same 10 BM samples from the competition cohort as the model. The results obtained from flow cytometry (FCM) were considered the established and accurate reference for comparison. For the identification of > 5% of blasts in the validation group, BMSNet (AUC 0.948) surpassed hematologists (AUC 0.929) but lagged behind pathologists (AUC 0.985). For the detection of over 20% of blasts, hematologists (AUC 0.981) and pathologists (AUC 0.980) showed similar but higher AUC values compared to BMSNet (AUC 0.942) [23].

In this study, the model presented showed great potential as a tool for hematologists to properly quantify blasts, which is essential in the diagnosis of MDS and other hematological malignancies. It was suggested by the researchers that well-trained hematologists should review the results of the AI interpretation before relying on them for patient decisions. Nevertheless, this would still save hematologists a lot of time in evaluating bone smears. One of the main drawbacks of this model, however, is that it was only internally validated and not externally validated. Moreover, the model was trained to only classify cells into 8 categories due to the difficulty of detecting intricate details that distinguish other cell types. Since this model required slide scanning, combining automatic slide scanners with an AI model would cut down the screening time for bone marrow samples dramatically [23].

Another issue that pertains to the diagnosis of MDS is its differentiation from AA and leukemia. It is important to rule out AA when diagnosing MDS because these two conditions share some similar clinical and hematological features, especially hypocellular MDS [35,36]. Since both hematological conditions result in cytopenia, they can sometimes be confused for one another. Current diagnostic methods include hematologic analysis, bone marrow biopsy, cytogenetics, and flow cytometry (FC). Pathological hematopoiesis is nonspecific and occurs in both states. Once thought to be dependable, cytogenetic abnormalities are no longer reliably unique to MDS. While FC has grown in popularity, its single marker usage and limitations in detecting erythroid malignancies make it difficult to diagnose MDS in general [37,38,39]. In addition to their similarities, MDS and AA are difficult to distinguish clinically due to the poor specificity of numerous indications.

To address this issue, a study by Wang et al. presented a deep learning model for the automatic diagnosis of MDS and the distinction between AA and AML based on BM smears [19]. The model was developed using a CNN and trained with data extracted from the American Society of Hematology (ASH) Image Bank, while external validation was performed using data from the clinic. Data from the ASH were randomly divided in a 7:3 ratio into training and testing datasets. Three different epochs were used for each model (30, 50, and 200). This determines the number of times the training set is presented to the learning model. The model had two output layers: whether the patient has MDS or not (two classifications) and whether they have AA, MDS, or AML (three classifications). The best model training effect was achieved with an outcome weight and epoch of 1:9 and 200, respectively. On external validation, the model exhibited high performance metrics in distinguishing MDS from non-MDS (AUC: 0.942, ACC: 0.921, SEN: 0.886, SPE: 0.938) and in distinguishing MDS, AA, and AML (AUC: 0.948, ACC: 0.915, SEN: 0.887, SPE: 0.929) [19]. Overall, the image-net pretrained model provided a convenient and accurate tool for clinicians to differentiate AA, MDS, and AML based on bone marrow smear images.

A similar model was also proposed by Wu, J. and colleagues that focused solely on the differential diagnosis of MDS from AA using decision tree ML models [22]. They developed multiple ML models, including SVM, LogR, a decision tree, and a BP network. Their models utilized data from peripheral blood counts, peripheral blood morphology, and bone marrow cell morphology from 130 patients with hypo-MDS and 156 patients with AA. These data were divided into 73% and 27% for the training and testing sets, respectively. Out of all the ML models utilized, the decision tree model outperformed all other models for the differentiation between MDS and AA with an AUC of 0.8, ACC of 0.805, SEN of 0.765, and SPE of 0.837 [22].

### 3.2. Diagnosis of MDS Using PBS

The conventional diagnosis of MDS from peripheral blood smears (PBS) presents its own set of challenges. PBS offer a snapshot of hematological abnormalities and can provide crucial insights into the diagnosis of MDS. However, similar to BM smears, the manual examination of PBS is time-consuming, subject to human error, and often requires experienced hematologists [40]. These challenges have paved the way for the application of AI techniques to enhance the accuracy, efficiency, and objectivity of MDS diagnosis using peripheral blood smears.

Multiple studies have shown that hypogranulated dysplastic neutrophils on PBS can provide valuable insights into the diagnosis of MDS [41,42,43,44]. However, it is sometimes challenging for pathologists to identify them on PBS. Hence, Acevedo and colleagues aimed to address the issue of identifying hypogranulated dysplastic neutrophils in peripheral blood by developing eight ML models labeled M1 to M8 using a CNN to undertake this task [24]. These models varied in architectural elements and training methodologies but were all trained for 20 epochs. The researchers established cut-off values for a granularity score to help the model distinguish between normal and dysplastic neutrophils, and they determined a threshold for identifying a minimum proportion of dysplastic neutrophils indicating a potential MDS diagnosis. The top five performing models were further trained for 100 epochs. Of these, the highest-performing model was M1. This model was internally validated, demonstrating high performance with an AUC value of 0.982, an ACC of 0.949, a SEN of 0.955, and a SPE of 0.943 [24]. Their work introduces an automated and objective method for identifying hypogranulated neutrophils, with potential application as an evaluation tool for MDS diagnosis within clinical laboratory workflows.

Another model was also proposed by Kimura et al. for automatic MDS differentiation from AA through a CNN utilizing PBS data [25]. They combined a CNN-powered DLS with the automatic detection and recognition of blood cells with an XGBoost decision-making system. Over 690,000 blood cell images from 3281 PBS were utilized in the training of their CNN model. Their model was able to classify 17 different blood cell types and their 97 morphological characteristics with an impressive SEN and SPE of 0.935 and 0.960, respectively. Their final model was able to distinguish MDS from AA utilizing PBS data with an AUC of 0.99, SEN of 0.962, SPE of 1.00, and overall ACC of 0.900. The limitations of their model included the adjunctive nature of the system, requiring additional diagnostic methods, and the need for clinical and genetic data for a definitive diagnosis. The study acknowledged the small sample size and single-center design, proposing future work to expand the dataset and enhance accuracy using serum biochemistry data [25].

A study by Zhu et al. aimed to evaluate the diagnostic performance of the Myelodysplastic Syndromes Complete Blood Count (MDS-CBC) score [26]. This is a score used clinically to exclude or suspect MDS in patients with cytopenia for unknown reasons at the time of identification. The authors sought to enhance MDS detection and reduce excessive smear reviews by incorporating the immature platelet fraction (IPF) into the MDS-CBC score. A total of 525 patients were included in the study, of which 168 had MDS. A random forest model was employed to identify the most effective predictors for MDS diagnosis. Notably, neutrophil structural dispersion (Ne-WX) and IPF emerged as the strongest predictors. They were then integrated into a Classification and Regression Trees (CART) model to refine the diagnostic accuracy of the current MDS-CBC score. A two-step approach was established, wherein patients with an MDS-CBC score ≤ 0.23 were classified as low-risk, and those exceeding this threshold were further stratified based on an IPF threshold of 3%. Results demonstrated the potential of the extended MDS-CBC score to enhance MDS diagnosis. The algorithm achieved a sensitivity of 84.5% and a specificity of 97.8%, with positive and negative predictive values of 94.7% and 93.1%, respectively [26].

The study leveraged machine learning techniques and included IPF as a novel parameter to enhance the MDS diagnosis by MDS-CBC score. By incorporating IPF into the model, the e-MDS-CBC score utilized the collective predictive power of the three myeloid lineages for MDS diagnosis. The application of random forest analysis and CART modeling allowed for the selection of key parameters and the formulation of decision trees suitable for laboratory middleware. However, the study also acknowledged certain limitations. The cohort consisted of individuals with suspected MDS, potentially impacting the algorithm’s performance in broader populations. Economic considerations were not extensively explored, and the cost-effectiveness of implementing IPF measurement for routine diagnosis requires further investigation. Additionally, the authors emphasized the importance of clinical judgment and the potential for slide review even in cases with low e-MDS-CBC scores, highlighting the complementary role of laboratory findings and clinical assessment.

### 3.3. Diagnosis of MDS Using FC

FC serves as a crucial tool in the diagnosis of MDS, aiding in the recognition of specific cellular attributes and counts that characterize this complex hematologic disorder. By enabling the precise analysis of individual cells, FC assists in identifying distinct markers and aberrant expression patterns that are indicative of MDS [45,46,47]. Despite its utility, the current utilization of FC faces challenges such as labor-intensive manual data interpretation, subjectivity in gating procedures, and a lack of standardized quantification, all of which hinder its efficiency and consistency in MDS diagnosis [45,48]. To overcome these limitations, AI emerges as a potential solution. AI offers the capacity to automate and optimize the analysis of high-dimensional flow cytometry data using advanced machine learning techniques. AI has the potential to enhance diagnostic accuracy, reduce variability, and uncover subtle cellular features that may hold diagnostic significance. Integrating AI into flow cytometry-based MDS diagnosis has the potential to revolutionize the field, addressing current limitations and providing a more efficient and precise approach to characterizing this challenging hematologic disorder.

Valentin Clichet et al. introduced an innovative approach combining AI with multiparametric FC to enhance MDS diagnosis and classification [27]. Their machine learning model employed an elasticnet algorithm applied to a cohort of 191 patients suspected of MDS. The research focused solely on flow cytometry parameters and utilized the Boruta algorithm for feature selection in the model. Granulocyte/lymphocyte SSC peak channel ratio, total hematogone ratio, percentage of CD34+ B-cell progenitors among all CD34+ cells, and the percentage of CD34+ myeloid progenitors were found to be the most important predictors for MDS diagnosis by the Boruta algorithm. The AI-assisted MDS prediction score (elasticnet model) demonstrates superior sensitivity to the existing Ogata score, maintaining excellent specificity. An external validation cohort of 89 patients confirms its high performance, with an AUC of 0.935. Notably, this model effectively diagnoses both high- and low-risk MDS, achieving 91.8% SEN and 92.5% SPE. Moreover, it reveals a progressive evolution of the prediction score from clonal hematopoiesis of indeterminate potential (CHIP) to high-risk MDS, implying a linear progression between these stages. Importantly, the AI-assisted prediction score significantly reduces misclassification rates, outperforming the Ogata score and establishing itself as a reliable diagnostic tool [27].

This study leverages AI to discriminate between MDS patients and non-MDS patients based on MFC profiles. The cohort encompasses patients from three distinct centers, ensuring the robustness and generalizability of the results. The diagnostic performance of the model was further confirmed using an external validation cohort, highlighting the model’s reliability and transferability. However, the flow cytometry data were acquired using different instruments, and the study acknowledges potential variability. Despite this, the AI-assisted model demonstrated consistent performance across the varied instruments, suggesting its widespread applicability. The model’s favorable attributes include speed, accessibility, and alignment with the Ogata score panel. In the context of cost-effectiveness, the AI-assisted prediction score offers a rapid and accurate approach for MDS diagnosis and stratification.

In another study by Carolien Duetz et al., a computational tool for FC diagnostics in suspected MDS was also developed and validated [28]. The study cohort consisted of 230 patients, including MDS patients and non-neoplastic cytopenia patients as age-matched controls. FC data were collected using a standardized panel of six tubes, and the preprocessing involved quality control and the exclusion of outliers. The diagnostic workflow incorporated the FlowSOM algorithm for cell population detection and a Random Forest ML classifier. The workflow was compared with expert-analyzed FC scores, such as the integrated flow cytometry score (iFS) and the Ogata score. The computational workflows outperformed these scores in terms of accuracy, objectivity, and time investment, with processing times reduced to less than 2 min per patient. In addition, a single-tube computational workflow was developed, which exhibited even higher SEN (97%) and SPE (95%) in the external validation cohort. Notably, the computational workflow revealed that certain cellular properties, particularly those of erythroid and myeloid progenitors, played a crucial role in diagnosing MDS patients. These properties were identified as the most relevant features for distinguishing between MDS and control cases [28].

The study demonstrated the advantages of the computational approach, including reduced processing time, cost-effectiveness, and enhanced diagnostic accuracy. The workflow’s performance was rigorously validated internally through cross-validation and externally using an independent cohort. Moreover, the study investigated different subgroups of MDS patients, such as those with excess blasts, and demonstrated consistent diagnostic accuracy. While the computational workflow holds great promise, the authors acknowledged certain limitations. The use of scatter parameters, although informative, posed challenges for standardization across different centers.

A different approach was taken by Maik Herbig et al., introducing a novel approach for diagnosing MDS using real-time deformability cytometry (RT-DC) combined with machine learning techniques [29]. Their study aimed to enhance MDS diagnosis by leveraging the quantitative image analysis capabilities of RT-DC and machine learning algorithms. RT-DC, an imaging FC, enables rapid acquisition of the morphological and mechanical properties of single cells. To assess the feasibility of this approach, BM biopsy samples from both healthy individuals and MDS patients were measured using RT-DC. Automated image analysis quantified seven features from each cell, capturing information related to cell size, mechanical properties, and porosity. A random forest model was trained using these features to distinguish between healthy and MDS samples. Internal validation of the model yielded compelling results with an AUC of 0.950, an ACC of 0.910, a SEN of 0.860, and a SPE of 1.000. The key features used for classification were those describing the width of cell size distribution, indicating that MDS samples exhibited narrower distributions compared to healthy ones [29]. This finding aligns with the WHO guidelines that consider cell size during MDS diagnosis [49].

Although the study presents a promising approach, several limitations and future directions were acknowledged. The sample size was relatively small, and the model’s generalization to a larger and more diverse MDS population requires further investigation. The current focus on HSCs should be expanded to include unsorted bone marrow to account for potential morphological differences resulting from mutated cells. Additionally, the technique’s effectiveness on fresh bone marrow samples should be explored.

Jeng-Lin Li et al. proposed an innovative automated algorithm for the diagnosis and classification of hematological malignancies, including MDS, based on deep phenotype representation [30]. The authors’ algorithm leverages a deep learning model to automatically classify minimal residual disease (MRD) into AML, MDS, and normal. The research utilizes a dataset retrospectively sourced from the National Taiwan University Hospital (NTUH), incorporating 2424 FC specimen samples. Each sample consisted of 11 tubes, each with a distinct channel–antibody pairing, facilitating measurement in six fluorescent channels. The raw cytometry data was initially transformed into a latent space using a per-tube autoencoder. Furthermore, the specimen-level representation was achieved through the Fisher-scoring vectorization approach, which combines generative modeling with discriminative power. A logistic regression model was utilized to perform four binary classifications (AML and MDS vs. Normal, AML vs. MDS, AML vs. Normal, and MDS vs. Normal). The model was subject to 5-fold cross-validation, where 20% of the dataset was used for training and 80% for testing. For the diagnosis of MDS (distinguishing MDS from normal), the model achieved an AUC of 0.956 with an accuracy of 0.960. For differentiating MDS from AML, the model achieved an AUC of 0.911 with an accuracy of 0.875 [30].

The significance of the research lies not only in its accuracy but also in its insights into disease classification. The authors emphasize that even with only half of the FC markers, the algorithm maintains high recognition accuracy, shedding light on the discriminability of existing markers. Moreover, the approach highlights the potential for reducing marker redundancy through computational methods. This novel algorithm consistently outperforms other representations across various classification tasks, emphasizing the importance of cell-level feature representation facilitated by autoencoder learning. While the findings of this study hold promise for advancing MRD classification, certain limitations warrant consideration. The observed discrepancies in classification accuracy between AML, MDS, and normal categories might stem from inherent complexities in categorizing MDS and potential data imbalances. Furthermore, the study’s focus on a specific dataset and markers necessitates further exploration to validate its applicability across broader contexts.

## 4. Discussion

The purpose of this study was to explore the diverse applications of ML algorithms in the diagnosis of MDS. In our investigation, we found a limited number of studies that have employed AI primarily for the diagnosis of MDS using PBS, BMS, and MFC data. The performance matrices of the ML models proposed by these studies demonstrated their great potential for the diagnosis of MDS, classifying patients at risk of MDS into low-risk or high-risk groups, and distinguishing MDS from its differentials like AA and AML. A significant proportion of the studies examined exhibited excellent predictive capabilities, with an AUC greater than 0.9. However, only three of the included studies performed external validation of their models. The collective evidence of these studies suggests that these models could serve as auxiliary tools to assist pathologists/hematologists in the diagnosis of MDS and offer a more cost- and time-effective diagnosis. However, these models have not been developed and tested extensively enough to replace the need for assessment of these samples by experienced hematologists/pathologists.

Given the extant research delving into the utilization of AI in the domain of MDS diagnosis, it is imperative to approach their outcomes with judicious circumspection. AI does have a more established role in other hematological conditions, such as ALL, where there has been extensive research [50]. We have also previously discussed the role of AI in other hematological diseases such as thrombocytopenia, sickle cell disease, chronic myeloid leukemia, and others [18,51,52,53,54]. Although there is still a long way to go before the diagnosis of hematological malignancies can be automated by AI, in its current state, AI can definitely assist hematologists and pathologists in diagnosis. As shown by some of the studies described above, AI has the potential to reduce the time, cost, and resources needed for MDS diagnosis and, hence, lead to earlier interventions in these patients and ultimately better patient outcomes.

A recurring observation seen in the examined studies was the lack of external validation of their models. Although these models performed exceptionally well on internal validation, it is sometimes misleading as the model might have plotted a random error in the sample and not true associations [55,56]. Although there are methodological approaches that limit such overfitting, they do not completely eliminate them [57,58]. Thus, a compelling imperative arises for the pursuit of external validation endeavors aimed at ascertaining the performance characteristics of these models when deployed across different samples and populations, independent of their original training datasets. Such an undertaking not only ensures the clinical applicability of these models but also safeguards against the undue limitation of their utility to a singular sample or population archetype.

Another issue commonly seen in these models is the utilization of a single source of data for the training of the ML models. The current diagnostic approach for MDS is multimodal. It typically involves a combination of clinical data, PBS, BMS, and FCM. In accordance with WHO guidelines, the diagnosis of MDS requires a combination of cytopenia with <20% blasts on PBS or BMS along with cytogenetic or morphological features of dysplasia [9]. Hence, AI models developed for the diagnosis of MDS should aim to combine this information to provide a more accurate diagnosis of MDS.

## 5. Conclusions

In conclusion, while the utilization of machine learning algorithms holds significant promise in the diagnosis of MDS, the current landscape is characterized by a limited yet encouraging body of research. These studies, employing various datasets, including PBS, BMS, and FC data, have exhibited noteworthy potential for accurately diagnosing and stratifying MDS patients. However, the absence of comprehensive external validation, coupled with the need for integrating diverse data sources representative of the multimodal diagnostic approach, underscores the imperative for cautious optimism. As AI continues its transformative journey in hematological disease diagnosis, its role as an assisting tool for pathologists and hematologists remains a compelling avenue, warranting further investigation and validation to unlock its full clinical potential in MDS management.

## Figures and Tables

**Figure 1 cancers-16-00065-f001:**
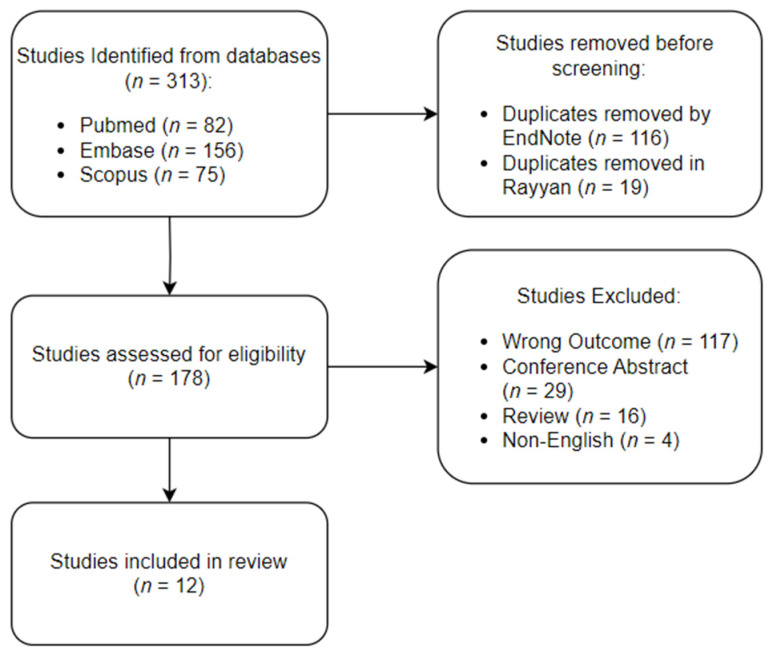
Schematic representation of the review process.

**Table 1 cancers-16-00065-t001:** Data extraction summary for the full-text articles included.

Study	Method	Outcome	Advantages	Disadvantages
Wang, M. et al. [19]	BMS	Diagnosing MDS and distinguishing it from AA and AML	Excellent performance metrics Internally and externally validated	Requires clinician assistance
Lee, N. et al. [20]	BMS	Detection of dysplastic erythrocytes, granulocytes, megakaryocytes, and blasts	Excellent performance metrics Competes with hematologists Detects dysplastic cells Internally validated	Does not quantify dysplasia Not externally validated
Mori, J. et al. [21]	BMS	Diagnosing MDS using hypogranulated dysplastic neutrophils	Excellent performance metrics Classifies dysplasia by severity Detection of dysplastic neutrophils Internally validated	Small sample size Not externally validated
Wu, J. et al. [22]	BMS and PBS	Diagnosing hypocellular MDS and distinguishing it from AA	Very good performance metrics Internally validated	Not externally validated Poor performance compared to other studies
Wu, Y. et al. [23]	BMS	Detection of elevated blasts to diagnose MDS	Quantifies dysplasia Internally validated	Not externally validated Only looks at blasts
Acevedo, A. et al. [24]	PBS	Detection of hypogranulated dysplastic neutrophils to diagnose MDS	Excellent performance metrics Internally validated Detects dysplastic neutrophils	Not externally validated
Kimura, K. et al. [25]	PBS	Diagnosing MDS and distinguishing it from AA	Excellent performance metrics Internally validated	Not externally validated
Zhu, J. et al. [26]	PBS	Diagnosing MDS using CBC and immature platelet fraction	Model outperforms current MDS-CBC scoring	Not externally validated
Clichet, V. et al. [27]	FC	Diagnosing MDS using MFC	Internally and externally validated Lower misclassification rates Excellent performance metrics	Lack of standardization of FC methodology
Duetz, C. et al. [28]	FC	Diagnosing MDS in suspected patients using FC	Excellent performance metrics Enhanced accuracy and reduced processing time Internally and externally validated	Lack of standardization of FC methodology
Herbig, M. et al. [29]	FC	Diagnosing MDS using RT-DC	Potential for efficient quantification Excellent performance metrics	Lack of standardization of FC methodology Small sample size Not externally validated
Li, J. L. et al. [30]	FC	Diagnosing MDS and distinguishing it from AML using FC	Excellent performance metrics	Lack of standardization of FC methodology Potential challenges in categorizing MDS due to data complexity Not externally validated

**Table 2 cancers-16-00065-t002:** Data sources and performance metrics for the best models in the included full-text articles.

Study	Data Source	Outcomes	Model Utilized	Validation	AUC	ACC	SEN	SPE
Wang, M. et al. [19]	American Society of Hematology image bank and Hospital BMS samples (AA, AML, MDS)	Diagnosing MDS	CNN	Internal	0.985	0.914	0.992	0.881
				External	0.942	0.921	0.886	0.938
		Distinguishing MDS from AA and AML	CNN	Internal	0.968	0.929	0.857	0.967
				External	0.948	0.915	0.887	0.929
Lee, N. et al. [20]	Hospital BMS (MDS and healthy controls)	Detecting dysplastic erythrocytes	CNN	Internal	0.972	0.988	0.790	0.992
		Detecting dysplastic granulocytes	CNN	Internal	0.996	0.993	0.900	0.999
		Detecting dysplastic megakaryocytes	CNN	Internal	0.971	0.931	0.899	0.948
		Detecting blasts	CNN	Internal	0.973	0.932	0.831	0.951
Mori, J. et al. [21]	Hospital BMS (MDS, “other hematological diseases”)	Diagnosing MDS using severe dysplasia (DG-3)	CNN	Internal	0.944	0.972	0.910	0.977
		Diagnosing MDS using dysplasia and severe dysplasia	CNN	Internal	0.921	0.982	0.852	0.989
Wu, J. et al. [22]	Hospital BMS and PBS (Hypo-MDS, AA)	Diagnosing hypocellular MDS and distinguishing it from AA	Decision tree	Internal	0.800	0.805	0.765	0.837
Wu, Y. et al. [23]	Hospital BMS (MDS, multiple myeloma, MPD, AA, lymphoma)	Detecting > 5% blasts	CNN: BMSnet	Internal	0.948	NR	NR	NR
Acevedo, A. et al. [24]	Hospital PBS samples (MDS and healthy controls)	Detecting hypogranulated dysplastic neutrophils	CNN: model M1	Internal	0.982	0.949	0.955	0.943
Kimura, K. et al. [25]	Hospital PBS data (MDS, MPN, AML, ALL, multiple myeloma, multiple lymphoma)	Diagnosing MDS and distinguishing it from AA	CNN with Xgboost	Internal	0.990	>0.900	0.962	1.000
Zhu, J. et al. [26]	Hospital PBS (MDS and non-MDS controls)	Diagnosing MDS	CART	Internal	NR	NR	0.845	0.978
Clichet, V. et al. [27]	Hospital MFC data (MDS)	Diagnosing MDS	Elasticnet (LinearR)	External	0.935	NR	0.918	0.925
Duetz, C. et al. [28]	Hospital FC data (MDS, healthy controls, non-neoplastic cytopenia)	Diagnosing MDS in suspected patients	Random forest	Internal	0.964	NR	0.850	0.950
				External	NR	NR	0.970	0.950
Herbig, M. et al. [29]	University Hospital RT-DC data (MDS, AML, CML, AA)	Predicting MDS	Random forest	Internal	0.950	0.910	0.860	1.000
Li, J. L. et al. [30]	Hospital FC data (AML, MDS, normal)	Classification of MDS vs. Normal	LogR using AGF-P	Internal	0.956	0.960	NR	NR
		Classification of MDS vs. AML	LogR using AGF-P	Internal	0.911	0.875	NR	NR

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
