# Peer review of "Integrating AI and ML in Myelodysplastic Syndrome Diagnosis: State-of-the-Art and Future Prospects"

_cancers, 2023, doi:10.3390/cancers16010065_

Round 1
Reviewer 1 Report
Comments and Suggestions for Authors
Any information about AI contribution in differentiation of MDS with excess of blasts(<20%) from myelofibrosis.
Reviewer 2 Report
Comments and Suggestions for Authors
The study by Elshoeibi et al is a review of the litterature in thield of AI as tool for diagnosis of MDS. The main question addressed by the paper is the state of the art of use of IA in diagnosis of MDS.
The study described advantages and limitations of each published AI tools.
The work is clear and documented.
A review of such area is very rare (AI in diagnosis of hematological malignancies). The originality is to analyse each published tool in term of advantages and limitations.
No other review deal with the subject, and the review itself is complete since most tool using AI are not yet published.
The references are appropriate, and the flow chart in the review is correctly described.
Comments on the Quality of English LanguageNo major comment.
Interesting and need minor editing of English.
Reviewer 3 Report
Comments and Suggestions for Authors Early diagnosis of MDS is of great importance to prevent it develoop into more aggressive disease and to improve current ways of diagnosing MDS. In this review, the authors summarized the state-of-the-art ML models for diagnosis of MDS. Major comments:- it would nice to include one column in table 1 to indicate if the method is using PBS, BMS or FC.
- since different methods use different dataset to train their models and validate their results, i think the authors should have a section about the dataset used in reviewed models. The model performance is somehow related the dataset used in the model. To make fair comparison of different models, it’s better to use the same dataset. I suggest the authors to include the dataset info in table 2 so the authors know if the results are directly comparable or not.
